# Surface Biofunctionalization of Gadolinium Phosphate Nanobunches for Boosting Osteogenesis/Chondrogenesis Differentiation

**DOI:** 10.3390/ijms24032032

**Published:** 2023-01-19

**Authors:** Zhongxing Cai, Ziyi Guo, Chaohui Yang, Fei Wang, Peibiao Zhang, Yu Wang, Min Guo, Zongliang Wang, Jing Huang, Long Zhang

**Affiliations:** 1School of Chemical Engineering, Changchun University of Technology, Changchun 130012, China; 2School of Advanced Institute of Materials Science, Changchun University of Technology, Changchun 130012, China; 3Key Laboratory of Polymer Ecomaterials, Changchun Institute of Applied Chemistry, Chinese Academy of Sciences, Changchun 130022, China

**Keywords:** surface modification, rare earth, biological evaluation, osteoblast differentiation, bone tissue engineering, MRI

## Abstract

In order to achieve smart biomedical micro/nanomaterials, promote interaction with biomolecules, improve osteogenic/chondrogenic differentiation, exhibit better dispersion in bone implants and ultimately maximize functionality, we innovatively and successfully designed and synthesized polymer PBLG-modified GdPO_4_·H_2_O nanobunches by hydroxylation, silylation and glutamylation processes. The effects of different feeding ratios on the surface coating of GdPO_4_·H_2_O with Si-OH, the grafting γ-aminopropyltriethoxysilane (APS) and the in situ ring-opening polymerization reaction of poly(g-benzyl-L-glutamate) (PBLG) were investigated, and the physical and chemical properties were characterized in detail. When GdPO_4_·H_2_O@SiO_2_–APS:NCA = 4:1, the PBLG-g-GdPO_4_·H_2_O grafting rate was 5.93%, with good stability and dispersion in degradable polymeric materials. However, the MRI imaging signal was sequentially weakened as the modification process proceeded. Despite this, the biological effects had surprising findings. All the modifiers at appropriate concentrations were biocompatible and biologically active and the biomacromolecules of COL I and COL II in particular were expressed at least 3 times higher in GdPO_4_·H_2_O@SiO_2_ compared to the PLGA. This indicates that the appropriate surface modification and functionalization of gadolinium-containing micro/nanomaterials can promote interaction with cells and encourage bone regeneration by regulating biomacromolecules and can be used in the field of biomedical materials.

## 1. Introduction

Biomaterials incorporating rare-earth micro/nanoparticles hold promise for multifunctional applications due to their excellent optical, electrical and magnetic properties, especially when used for drug delivery, bioimaging and targeted therapies [1,2,3,4]. Rare earth gadolinium-containing micro/nano materials not only have luminescent properties, but also magnetic resonance tracking properties, making them easy to form multifunctional materials [5,6,7]. In particular, in the field of bone tissue engineering, due to their unique biocompatibility, antimicrobial, antioxidant and anti-inflammatory properties, a range of smart nanobiotics with improvements has been developed for current investigations to address the challenges and problems faced in the field of bone tissue engineering and implantology [8].

In addition to calcium phosphates (CaPs), which bring good bone mineralization activity to bone replacement materials [9], nanoparticles containing gadolinium, europium, SiO_2_, etc., will form a protein corona [10], a key mediator of nanoparticle–cell interactions, as they can still interact with cells, will facilitate osteogenesis and angiogenesis by modulating inflammatory responses [11,12,13], or promote bone regeneration by targeting key signaling pathways [14,15,16,17,18,19], using material composites or surface modifications that play a great role in bone repair [20]. However, due to the strong coordination of rare earth ions on exposed surfaces, these micro/nanoparticles are easily aggregated in polymer systems when used in degradable polymer bone implants. Therefore, research on the surface modification of gadolinium-containing micro/nano materials to ensure their properties such as luminescence and MRI tracing and to enhance their biocompatibility and bioactivity after functionalization is imminent.

Surface modification is used as an effective method to ensure high dispersion and stability in heterogeneous systems and to avoid agglomeration of inorganic micro/nanoparticles. In addition, the use of a suitable surface modification to make inorganic micro/nanoparticles biocompatible and bioactive and to exploit the multifunctionality of rare earth micro/nanomaterials is a current research hotspot [21,22]. Among the different strategies, amphiphilic polymers, lipids, silica shells, ligand exchange, ligand oxidation, biopolymer polydopamine coating, etc., surface alkylation is a more classical approach for the surface modification of micro/nanoparticles [23,24,25,26]. Prior to this, TEOS (ethylene orthosilicate) was chosen to cover the surface of the micro/nanoparticles to form an amorphous silica cladding layer if there were fewer hydroxyl groups on the surface of the micro/nanoparticles. With this type of covering, a higher water solubility and a better photostability and biocompatibility can be obtained, and more importantly, conditions can be created for increasing the amino or carboxyl groups on the surface of the rare earth micro/nanoparticles, making them suitable for a wider range of biological applications [24]. Furthermore, as silane coupling agents are flexible and can carry groups such as carboxyl, amino and amide groups, more research has been carried out using silane coupling agents to modify the surface of the particles. For example, γ-aminopropyltriethoxysilane (APS)-modified nanoparticles have the advantage of water solubility and a high density of amino groups (-NH_2_) [27], which gives the modified magnetite and cerium oxide nanoparticles good dispersion and stability in water [28]. Moreover, it is certain that the PBLG-modified nanoparticles have a positive effect on bone differentiation and proliferation [29]. Nevertheless, comparative studies of the physicochemical and biological properties of the functionalized intermediates and products synthesized in each step of the reaction, such as hydroxylation, silylation and ammonification, exist but are not adequate and detailed.

Therefore, in this paper, GdPO_4_·H_2_O@SiO_2_, GdPO_4_·H_2_O@SiO_2_–APS and PBLG-g-GdPO_4_·H_2_O were designed and synthesized using surface-modified functionalized GdPO_4_·H_2_O nanobunches, with the aim of experimentally screening biofunctionalized gadolinium-containing nanomaterials with good dispersion and stability, a certain degree of paramagnetic MR imaging ability and facilitating osteogenic/chondrogenic differentiation. The structure-to-function conformational relationships were investigated by adjusting the ratios of GdPO_4_·H_2_O to TEOS and GdPO_4_·H_2_O@SiO_2_–APS to NCA monomers. The encapsulated morphology and elemental composition were observed by SEM, TEM and EDS, the changes in crystal structure were characterized by XRD, the chemical structure of the samples was studied by FT-IR and the grafting rate of the products was calculated by TG. The stability and dispersion of the samples in chloroform and PLGA were investigated by sedimentation experiments as well as AFM characterization, and the T_1_-weighted MRI signals of the different modifications were compared using a 1.5 T MRI system. Cytotoxicity and biocompatibility were also evaluated by in vitro cellular assays such as cytotoxicity, cell adhesion and cell proliferation, and the osteogenic/chondrogenic activity and mineralization capacity of the different modifications were investigated by alkaline phosphatase activity assay, type I and type II collagen gene expression and calcium mineralization deposition experiment The possible mechanisms of the effect of functional modifiers of gadolinium phosphate nanobunches on osteogenic/chondrogenic activity were further investigated. The present study demonstrates different modification processes for successive GdPO_4_·H_2_O nanobunches. As the modification process proceeds, the functional variability of each modifier begins to manifest, laying the foundation for the functionalization of gadolinium-containing micro/nanomaterials in the field of bone tissue engineering and biomedical materials.

## 2. Results and Discussion

In previous work, good progress was made in investigating the effect of different gadolinium phosphate doping concentrations on the in vitro/in vivo tracer properties of bone implant composites and their in vitro/in vivo biological effects [18]. To further regulate the osteogenesis/cartilage induction activity, improve the dispersion of rare earth micro/nanoparticles in the matrix of bone implants, and investigate the changes in MRI tracing properties, GdPO_4_·H_2_O@SiO_2_, GdPO_4_·H_2_O@SiO_2_–APS and PBLG-g-GdPO_4_·H_2_O were obtained from different modification processes by a silica coating on GdPO_4_·H_2_O, silane coupling agent premodification and polyglutamic acid modification, as is shown in Figure 1. The surface modification and functionalization processes were based on a published method [30]. Firstly, we characterized the physicochemical properties of the synthesized intermediates and modification products (SEM, TEM, XRD, FT-IR, TG, sedimentation experiments, AFM analysis) in order to determine the optimal surface modification conditions. Secondly, we investigated the MRI tracer properties of each modifier to explore the effects of different degrees of surface modification and different grafting rate products on MRI T_1_ imaging. Finally, we performed biocompatibility and bioactivity evaluations such as cytotoxicity, cell adhesion, cell proliferation, ALP activity, gene expression of COL I and COL II and calcium mineralization deposition assays to ascertain the effect of the degree of functionalization of surface modifications on osteogenic/chondrogenic induction activity. The whole process is not only a further exploration of inorganic gadolinium-containing micro/nanomaterials for applications in bone tissue engineering, but also a solution to the current clinical problems faced by biodegradable orthopedic implant devices.

### 2.1. Design and Synthesis of PBLG-g-GdPO_4_·H_2_O

#### 2.1.1. Sufficient Suitable Encapsulation of SiO_2_ onto the Surface of GdPO_4_·H_2_O Nanobunches

To investigate the effect of the feeding ratio on the materials, GdPO_4_·H_2_O@SiO_2_ core-shell nanobunches were designed and synthesized in the ratios of m_GdPO4·H2O_:V_TEOS_ (mg/mL) = 1:8, 4:5, 1:1, 5:4, 5:3 and 5:2. Figure 1 shows the morphology of GdPO_4_·H_2_O@SiO_2_. According to previous reports in the literature, uncoated gadolinium phosphates are bundles with smooth surfaces [18]. However, there are many small nanoparticles around the bundles in Figure 1A, indicating that a ratio of 1:8 resulted in an excess of TEOS and the formation of too many silica nanoparticles around gadolinium phosphate. When the feed ratio of GdPO_4_·H_2_O and TEOS was greater than or equal to 4:5, the excess silica particles around the nanobunches in Figure 1B–F were significantly reduced or even absent. At the same time, the surface of the coated sample was rough compared to the previously synthesized pure GdPO_4_·H_2_O. To further determine the core–shell structure, TEM analysis was performed on the samples without excess SiO_2_ nanoparticles. A distinct core–shell structure can be observed by TEM in Figure 2, and with a gradual increase in the feed ratio, the SiO_2_ coating became progressively thinner, gradually decreasing from 9.94 nm to 5.16 nm, which further confirms the successful encapsulation of SiO_2_ onto the surface of the GdPO_4_·H_2_O nanobunches.

To ensure that the surface of gadolinium phosphate was completely covered by SiO_2_, zeta potential analysis was carried out on GdPO_4_·H_2_O@SiO_2_ synthesized without excess SiO_2_ particles. As shown in Table 1, when the feed ratios of GdPO_4_·H_2_O to TEOS were 4:5, 1:1, 5:4, 5:3 and 5:2, the zeta potentials of the resulting GdPO_4_·H_2_O@SiO_2_ were −26.52 mV, −30.69 mV, −16.98 mV, −16.96 mV and −12.71 mV, respectively. It shows that as the amount of TEOS decreased, the negative charge on the surface of GdPO_4_·H_2_O showed a trend of first increasing and then decreasing. At the ratio of 4:5, the surface of GdPO_4_·H_2_O was covered with more SiO_2_, and the SiO_2_ in the outer layer covered the SiO_2_ in the inner layer of the shell, so that part of the Si-OH was blocked, thus reducing the negative charge (−26.52 mV). The surface of GdPO_4_·H_2_O was just completely encapsulated by SiO_2_ when the feed ratio was 1:1, and all Si-OH was exposed on the surface, so the surface had the most negative charge and the lowest zeta potential value (−30.69 mV). With the ratios of 5:4–5:2, the surface of GdPO_4_·H_2_O was coated with less SiO_2_, and the Si-OH was reduced; hence, the negative charge gradually diminished, and the zeta potential gradually rose (−16.98 mV–−12.71 mV). The lowest zeta potential value (−30.69 mV) was achieved when the feed ratio of GdPO_4_·H_2_O to TEOS was 1:1 (mg/mL), indicating that the optimum amount of TEOS could be achieved at this ratio. At this point, neither too many SiO_2_ nanoparticles were produced nor too little SiO_2_ made the GdPO_4_·H_2_O nanobunches incompletely encapsulated. More importantly, sufficiently suitable encapsulation was conducive to producing the lowest surface charge and more silicon hydroxyl groups on the surface of GdPO_4_·H_2_O, which was beneficial for the next modification step.

Based on the above discussion, EDS analysis and elemental distribution scans were performed for GdPO_4_·H_2_O@SiO_2_ prepared with m_GdPO4_·_H2O_:V_TEOS_ (mg/mL) = 1:1. Figure 3A shows the corresponding scanned site. As shown in Figure 3B, the present of the elements Gd, Si, P and O in the GdPO_4_·H_2_O@SiO_2_ sample was demonstrated due to the matching of the detected peaks (blue area) and the elemental identification positions (red line). The detection of C element may be the result of a small amount of adsorption of carbonaceous compounds from the air by the sample. Furthermore, according to the elemental distribution scans in Figure 3C, the elements Gd (green and blue) and P (yellow) were mainly concentrated in the interior, while the element Si (orange) was mainly concentrated in the exterior of the core–shell structure. Since both GdPO_4_·H_2_O and SiO_2_ contain oxygen, the O element (red) was present throughout the structure. All this confirms the successful synthesis of the GdPO_4_·H_2_O@SiO_2_ core–shell structure with more Si-OH groups.

#### 2.1.2. Further Surface Modification of GdPO_4_·H_2_O@SiO_2_ with Polymer APS and PBLG

GdPO_4_·H_2_O@SiO_2_–APS and PBLG-g-GdPO_4_·H_2_O@SiO_2_–APS (PBLG-g-GdPO_4_·H_2_O) were synthesized by successive silylation with APS on the surface of GdPO_4_·H_2_O@SiO_2_, followed by glutamylation with PBLG. The successful synthesis of GdPO_4_·H_2_O@SiO_2_ provided sufficient hydroxyl groups on the surface of GdPO_4_·H_2_O. Fixed synthesis ratios GdPO_4_·H_2_O@SiO_2_:APS = 1:0.22 were used to obtain GdPO_4_·H_2_O@SiO_2_–APS by grafting the silane coupling agent APS. Glutamate-modified PBLG-g-GdPO_4_·H_2_O nanoparticles were obtained at different ratios of GdPO_4_·H_2_O-APS and NCA polyglutamate monomer. The specific reaction ratios are shown in Table 2.

XRD analysis of GdPO_4_·H_2_O@SiO_2_ and PBLG-g-GdPO_4_·H_2_O obtained with different feed ratios of GdPO_4_·H_2_O@SiO_2_–APS:NCA = 1:0.125, 1:0.25, 1:0.5 and 1:1 is shown in Figure 4. The XRD patterns of GdPO_4·_H_2_O@SiO_2_ and all PBLG-g-GdPO_4_·H_2_O products had diffraction peaks at 20.4°, 25.7°, 29.8°, 32.0°, 42.2° and 49.5° corresponding to the (101), (110), (200), (102), (211) and (212) crystal planes, in agreement with the standard card GdPO_4_·H_2_O (JCPDS 39-0232), indicating that modification with TEOS and PBLG did not alter the core phase of GdPO_4_·H_2_O [18,31]. By increasing the NCA monomer synthesized in Stage 2 of Figure 1 (L-glutamic acid γ-benzyl ester-N-carbonyl-lactam anhydride), the diffraction peaks of PBLG-g-GdPO_4_·H_2_O weakened as a result of the increasing grafting of PBLG by decreasing the crystallinity. Although there was no change in the above XRD peak shifts, the amorphous broad peaks appeared at the ratios of GdPO_4_·H_2_O@SiO_2_–APS:NCA = 1:0.5 (D) and 1:1 (E). There was no significant difference in XRD for GdPO_4_·H_2_O@SiO_2_ (A) and PBLG-g-GdPO_4_·H_2_O with the ratios of GdPO_4_·H_2_O@SiO_2_–APS:NCA = 1:0.125 (B), 1:0.25 (C). This was mainly due to the fact that with the amorphous structure and thinner shell layer SiO_2_ encapsulation, no characteristic peak of the silicon shell was observed [32]. Moreover, the fewer NCA involved in the grafting reaction the less PBLG will be grafted to the surface of GdPO_4_·H_2_O@SiO_2_, producing a thinner surface modification layer and therefore have no affect on the XRD. In contrast, when GdPO_4_·H_2_O@SiO_2_–APS:NCA = 1:0.5 (D) and 1:1 (E), the silane coupling agent modification and the involvement of more NCA monomers in the polymerization reaction led to more polyglutamic acid grafting onto the GdPO_4_·H_2_O@SiO_2_ surface, resulting in a thicker modified layer and ultimately an amorphous broad peak of the polymer in XRD. No new crystalline peaks appeared in XRD, no significant changes in peak positions, and only new PBLG amorphous peaks appeared as the concentration of polyglutamic acid monomer (NCA) increased. These results indicate that the polymerization reaction occurred only on the surface of the gadolinium phosphate nanocrystals, without changing the crystal phase of GdPO_4_·H_2_O [30,33].

In order to further confirm the successful modification onto the surface of GdPO_4_·H_2_O, the FTIR spectra were investigated, as shown in Figure 5. The two bands at 3478 cm^−1^ and 1618 cm^−1^ in Figure 5A are attributed to the OH absorption peaks. The absorption peaks near 1075 cm^−1^, 800 cm^−1^ and 464 cm^−1^ are the Si-O-Si asymmetric stretching, symmetric stretching and bending vibration peaks, respectively [34]. The absorption peaks at 622 cm^−1^ and 545 cm^−1^ were caused by the O-P-O bending vibration and represent the characteristic absorption of PO_4_^3−^. Thus, the amorphous SiO_2_ shell coating on the surface of GdPO_4_·H_2_O nanobunches was confirmed in terms of chemical composition [32]. With silane coupling agent (APS) pre-modification and further polyglutamic acid grafting, the absorption peaks at 1549 cm^−1^, 1654 cm^−1^ and 1737 cm^−1^ in Figure 5B–E were progressively enhanced with increasing ratios of GdPO_4_·H_2_O@SiO_2_–APS:NCA = 1:0.125 (B), 1:0.25 (C), 1:0.5 (D) and 1:1 (E) as they are attributed to the stretching vibrational absorption peak of amide N-H, amide C=O and carboxy C=O, respectively, demonstrating the presence of PBLG chains on the surface of GdPO_4_·H_2_O [33]. These results further confirm the successful preparation from GdPO_4_·H_2_O, GdPO_4_·H_2_O@SiO_2_ and GdPO_4_·H_2_O@SiO_2_–APS to PBLG-g-GdPO_4_·H_2_O.

On the basis of the TG curves in Figure 6, the grafting rates of PBLG were calculated [30], and the detailed results are shown in Table 2. By adjusting the ratios of GdPO_4_·H_2_O@SiO_2_–APS:NCA = 1:0.125, 1:0.25, 1:0.5 and 1:1, the grafting rates were 4.14%, 5.93%, 13.72% and 15.93%, respectively. The grafting rate gradually rose with the increase in NCA monomer, leading to more PBLG grafting onto the surface of GdPO_4_·H_2_O@SiO_2_–APS, thereby preparing PBLG-g-GdPO_4_·H_2_O with different grafting rates.

### 2.2. Dispersion and Stability

To observe the dispersion and stability of the different modifiers in the solvent, the synthesized samples were dispersed in chloroform solution [29]. After ultrasonic dispersion, the samples were monitored with a conventional camera to observe the sedimentation, as shown in Figure 7. GdPO_4_·H_2_O@SiO_2_ without PBLG grafting precipitated immediately, and PBLG-g-GdPO_4_·H_2_O with 13.72%; (D) and 15.93% (E) grafting precipitated after 10 s. When resting for 5 min, a small amount of precipitation occurred for PBLG-g-GdPO_4_·H_2_O with a grafting rate of 4.141%; (B), and 5.926% (C) exhibited excellent colloidal stability for a longer period of time (15 min). Without grafting or at lower grafting rates (0% and 4.14%), the PBLG chains were less able to maintain the stability of the gadolinium phosphate nanoparticles in chloroform solution and were more prone to settling. Despite the relatively high grafting rate of PBLG-g-GdPO_4_·H_2_O (13.72% and 15.93%), more NCA monomers formed longer PBLG polymer chains attached to the surface of GdPO_4_·H_2_O@SiO_2_, causing surface modification supersaturation, low colloidal stability and short settling times in chloroform. This was further confirmed by the previous XRD (Figure 4) and FT-IR (Figure 5) analyses. Thus, PBLG-g-GdPO_4_·H_2_O with a grafting rate of 5.926% maintained good stability of the micro/nanoparticles in chloroform solution.

To further confirm that the grafting products could be uniformly dispersed in the degradable polymer PLGA, the surface of the PBLG-g-GdPO_4_·H_2_O/PLGA film was observed by atomic force microscopy (AFM), as shown in Figure 8. PBLG-g-GdPO_4_·H_2_O with different grafting rates all had good dispersion and could be uniformly dispersed in the degradable polymer material PLGA.

### 2.3. T_1_-Weighted MR Imaging of Modifiers in Biomedical Polymer Material of PLGA

Scaffold samples of PLGA (A), GdPO_4_·H_2_O/PLGA (B), GdPO_4_·H_2_O@SiO_2_/PLGA (C), GdPO_4_·H_2_O@SiO_2_–APS/PLGA (D) and PBLG-g-GdPO_4_·H_2_O/PLGA (E-G) of the control, modification intermediates, and various grafting rate products were placed into a 1.5 T MRI instrument for monitoring. From the T_1_-weighted imaging (left) and T_1_-parametric map (right) images in Figure 9, it was found that GdPO_4_·H_2_O/PLGA had the strongest MRI signal as the surface modification and functionalization process proceeded, followed by GdPO_4_·H_2_O@SiO_2_/PLGA. GdPO_4_·H_2_O@SiO_2_–APS/PLGA (D) and PBLG-g-GdPO_4_·H_2_O/PLGA (E-H) had significantly weaker MRI signal intensities. The higher the grafting rate was, the weaker the T_1_-weighted MRI signal of PBLG-g-GdPO_4_·H_2_O/PLGA. In the preparation of inorganic‒organic bone implants, the solvent NMP was replaced by water, while water molecular channels were formed in the degradable polymer of PLGA. Gadolinium can chirp near hydrogen protons during MRI monitoring, but after functional modification, the thick coating on GdPO_4_·H_2_O inhibits the interaction of gadolinium with surrounding hydrogen protons and produces the shielding effect on H^+^, resulting in a weaker MRI T_1_-weighted imaging signal. While many studies have been conducted on the surface modification and functionalization of inorganic rare earth micro/nanoparticles [35,36], it is relatively rare for this paper to investigate the effects of different modification process products on the MRI T_1_-weighted imaging signal of inorganic gadolinium-containing micro/nanoparticles, which also provides a reference for later researchers to make reasonable use of the surface modification process to prepare MRI tracing-related functionalized materials. As a paramagnetic contrast agent for MRI of bone tissue, a more suitable method for enhancing the development of gadolinium phosphate needs to be found in the future.

### 2.4. Cellular Response

#### 2.4.1. Cytotoxicity and Biocompatibility

A systematic in vitro experimental evaluation of the materials was carried out to investigate the biocompatibility and bioactivity of the modification intermediates and modification products. The cytotoxicity experiments are shown in Figure 10. For GdPO_4_·H_2_O/PLGA, GdPO_4_·H_2_O@SiO_2_–APS/PLGA and PBLG-g-GdPO_4_·H_2_O/PLGA, the cell viability of the material leachate increased progressively with increasing dilution, suggesting a possible “low-promoting, high-inhibiting” hormesis effect [37], with no significant cytotoxic effect on MC3T3-E1 cells at the appropriate concentration. However, the survival rate of GdPO_4_·H_2_O@SiO_2_/PLGA decreased step by step, indicating that the SiO_2_ coating had a positive effect on promoting cell survival with increasing concentrations in this range. Notably, only the GdPO_4_·H_2_O@SiO_2_–APS/PLGA leachate showed a cell survival rate of 73.4% without dilution, which was lower than 75%, indicating a slight and low toxic effect due to the presence of a relatively high amount of silane coupling agent APS [38,39,40]. Other groups and cell experiments with dilutions of the extract showed no significant cytotoxicity to MC3T3-E1 cells. The above results predict good biocompatibility of all surface modifiers.

Moreover, the adhesion behavior of MC3T3-E1 cells was observed by FITC staining, as displayed in Figure 11. The microscopic images show that at 6 h of cell culture, a small number of cells began to adhere to the surface of the material, and the cells were mostly round or spindle shaped. At 12 h of incubation, the cells began to stretch and change to a polygonal shape. At 24 h of incubation, the number of cells began to increase, and the cell bodies were fully expanded and grew in colonies. There were no significant differences in cell numbers and cell morphology between MC3T3-E1 cells cultured for 6 h, 12 h and 24 h, suggesting that each modifier had no significant effect on cell adhesion of MC3T3-E1 in PLGA.

The rate of cell proliferation is the first important indicator of osteogenic differentiation, and when this phenotypic shift occurs, the level of proliferation will begin to plateau. To further confirm the biocompatibility of the material, the cell proliferation assay was carried out, as shown in Figure 12. There was no significant difference in the absorbance (OD) values of the cells cultured on the surface of each material when MC3T3-E1 cells were cultured for 1 day. As the culture time was extended to 3 days, the OD values detected in each group increased, indicating an increase in cell numbers. At this time, the OD values of the GdPO_4_·H_2_O/PLGA, GdPO_4_·H_2_O@SiO_2_/PLGA, GdPO_4_·H_2_O@SiO_2_–APS/PLGA and PBLG-g-GdPO_4_·H_2_O/PLGA groups were all higher than those of PLGA, especially the OD values of PBLG-g-GdPO_4_·H_2_O/PLGA cultured cells, which were the highest, with significant differences (*p* < 0.05). This indicates that the modifiers had good biocompatibility, which may have been the consequence of the positive biocompatibility and biological effects of GdPO_4_·H_2_O, SiO_2_, APS and polyglutamic acid in PBLG [32,41,42,43,44]. At 7 days of incubation, the OD values of the groups were similar, with only the GdPO_4_·H_2_O@SiO_2_/PLGA and PBLG-g-GdPO_4_·H_2_O/PLGA groups having slightly higher OD values than the glass, GdPO_4_·H_2_O/PLGA and GdPO_4_·H_2_O@SiO_2_–APS/PLGA groups, but there was no significant difference (*p* > 0.05). In the above cell experiments, it was demonstrated that the surface modification of GdPO_4_·H_2_O resulted in low toxicity of the modifiers to MC3T3-E1 cells in the experimental concentration range and in PLGA, with no significant inhibition of cell adhesion behavior and cell proliferation, but rather a pro-cell proliferation effect at 3 days of culture (notably PBLG-g-GdPO_4_·H_2_O), indicating that the modifiers were all biocompatible. This may have been due to the interaction of PBLG with adhesion proteins in the extracellular matrix, and integrins on the cell surface contributing to good cell proliferation [44].

#### 2.4.2. Osteogenic/Chondrogenic Induction Activity

Alkaline phosphatase (ALP), the metalloenzyme that catalyzes the nucleation of inorganic phosphates, is essential for the mineralization of hard tissues and is an early marker of the development of cells towards the differentiation pathway [45,46]. The ALP activities of MC3T3-E1 cells cultured on different materials for 7 and 14 days are shown in Figure 13. At 7 days of culture, the ALP activity of the GdPO_4_·H_2_O@SiO_2_/PLGA group was similar to that of the glass and higher than that of the GdPO_4_·H_2_O/PLGA, GdPO_4_·H_2_O@SiO_2_–APS/PLGA and PBLG-g-GdPO_4_·H_2_O/PLGA groups, with a significant difference (*p* < 0.05), indicating that SiO_2_-coated GdPO_4_·H_2_O facilitated ALP secretion by MC3T3-E1 cells. At 14 days of culture, the ALP activity of the gadolinium-containing group was higher than that of both the glass and PLGA groups, especially in the GdPO_4_·H_2_O@SiO_2_–APS/PLGA and PBLG-g-GdPO_4_·H_2_O/PLGA groups, with a significant difference (*p* < 0.05), indicating that the surface-modified functionalized intermediate GdPO_4_·H_2_O@SiO_2_–APS and the product PBLG-g-GdPO_4_·H_2_O promoted ALP secretion at 14 days of cell culture. The peptide PBLG itself stimulates cell adhesion and the APS also has a strong interaction with cells, which in turn facilitates further cellular processes such as proliferation and differentiation [44].

Collagen type I (COL I) is the most abundant component of the organic matrix in bone tissue with a length of 300 nm and a width of 1.1–1.5 nm [45]. Collagen type II (COL II) is a cartilaginous ECM molecule found mainly in cartilage and developing bone and is an important regulator of osteogenesis in BMSCs, with research showing that COL II is more effective in promoting osteogenesis in BMSCs [47]. To assess the effect of functionalized materials on the COL I and COL II of osteoblasts, the gene expression levels of MC3T3-E1 cells were analyzed by qRT‒PCR during growth and differentiation. The results are shown in Figure 14. At 7 and 14 days of culture, the COL I expression levels of MC3T3-E1 cells in GdPO_4_·H_2_O/PLGA, GdPO_4_·H_2_O@SiO_2_/PLGA, GdPO_4_·H_2_O@SiO_2_–APS/PLGA and PBLG-g-GdPO_4_·H_2_O/PLGA were higher than those of the glass and PLGA groups. The highest COL I expression was found in the GdPO_4_·H_2_O@SiO_2_/PLGA group and the expression level was up to 11.6 and 4.8 times higher than that of the PLGA group at 7 and 14 days, respectively. This indicates that due to the presence of SiO_2_, the GdPO_4_·H_2_O@SiO_2_/PLGA composite promoted the high expression of COL I during the proliferation and differentiation phase of the cells. For COL II expression, the expression level of GdPO_4_·H_2_O@SiO_2_/PLGA and GdPO_4_·H_2_O@SiO_2_–APS/PLGA was higher than that of the other material groups at 7 days, and there was a higher expression of GdPO_4_·H_2_O@SiO_2_/PLGA, GdPO_4_·H_2_O@SiO_2_–APS/PLGA and PBLG-g-GdPO_4_·H_2_O/PLGA at 14 days of culture. It is pretty obvious that the GdPO_4_·H_2_O@SiO_2_/PLGA group likewise showed the highest expression level with significant difference than the other composites. The COL II expression levels in the GdPO_4_·H_2_O@SiO_2_/PLGA group were up to 11.8 and 3.1 times higher than that in the PLGA group at 7 and 14 days, respectively. The above indicates that the modification of the SiO_2_ layer significantly promoted the expression of COL I and COL II molecules involved in bone reconstruction. Owing to the encapsulation of SiO_2_ on the GdPO_4_·H_2_O surface, the GdPO_4_·H_2_O@SiO_2_ will expose several negative charges or Si-OH and gradually release Si ions during the cell proliferation and differentiation phase. Simultaneously, SiO_2_/Si is known as a “biological cross-linker” and plays a particularly important role in linking proteoglycans and collagen, which is essential for biomineralization and collagen synthesis [48,49,50]. As early as 1970, Si was identified as a key element involved in tissue mineralization [51]. Si ions are strongly involved in the initial stages of biomineralization and a reduced intake of Si leads to marked defects in the growth of bone and cartilage [52]. Therefore, in this study, SiO_2_ and GdPO_4_·H_2_O may have acted synergistically to promote the high expression of COL I and COL II, thus contributing to the promotion of osteogenic/chondrogenic activity of the bone implant material. Of course, due to the activity of the substrate and the functionalization of the active modifiers, the up-regulation of the COL I and COL II genes by them or by their combined action at different incubation times should also be acknowledged.

After osteogenic differentiation, the cells begin to secrete a mineral matrix, known as the mineralization phase. The qualitative and quantitative assessments of mineral deposits were determined, as shown in Figure 15 and Figure 16. Calcium mineralization deposition of the cells was assessed when MC3T3-E1 cells were cultured on PLGA, GdPO_4_·H_2_O/PLGA, GdPO_4_·H_2_O@SiO_2_/PLGA, GdPO_4_·H_2_O@SiO_2_–APS/PLGA and PBLG-g-GdPO_4_·H_2_O/PLGA for 3 weeks. Upon alizarin red (ARS) staining, red calcium nodules appeared on the substrate of each material group, indicating that MC3T3-E1 cells secreted calcium ions and produced mineralized deposits when cultured for 3 weeks. After quantitative analysis, as shown in Figure 16, the PLGA, GdPO_4_·H_2_O/PLGA, GdPO_4_·H_2_O@SiO_2_/PLGA and PBLG-g-GdPO_4_·H_2_O/PLGA groups had the higher calcium content, with significant differences compared to GdPO_4_·H_2_O@SiO_2_–APS/PLGA group (*p* < 0.05). Of all modifiers, GdPO_4_·H_2_O@SiO_2_/PLGA had the highest calcium content, followed by group GdPO_4_·H_2_O/PLGA, possibly related to the previous finding of high expression of the COL I and COL II in GdPO_4_·H_2_O@SiO_2_. Studies have shown that COL I/COL II can promote calcium mineralization deposition through the unique role of integrin a2b1-FAK-JNK signaling in regulating osteogenic differentiation of BMSC [47]. Gd promotes the expression levels of ALP, COL-1 and Runx2 in conjunction with mesoporous SiO_2_ in chitosan scaffolds and subsequent osteogenic mineralization, upregulates p-GSK3β and β-catenin, important marker proteins of the Wnt/β-catenin signaling pathway [14]. Although the mineralization results in this experiment were not as pronounced, especially in the PBLG-g-GdPO_4_·H_2_O group, the results did not follow the same trend as the data already reported [44], presumably due to the lower level of introduced modifiers in the scaffold (0.4%, Gd/PLGA, *w*/*w*). Perhaps the advantages of mineralization would be better demonstrated by increasing the amount of PBLG-g-GdPO_4_·H_2_O.

### 2.5. Possible Molecular Mechanism

Osteogenesis is a complex physiological process that takes 4–6 months to complete skeletal modelling or remodeling. During ossification, multiple biomolecular signaling proteins (e.g., BMP, PDGF, FGF, TGF-β, Wnt, Hedgehog, etc.) are involved in regulating bone resorption and mineralization. In simulating natural bone and promoting osteogenesis in bone implants, the organic (matrix phase) and inorganic (reinforcing phase) phases, as smart and multifunctional materials, each have their own role to play and ‘look after’ each other to promote the osteogenic differentiation and mineralization process and neither is dispensable [45]. In this work, a variety of modifications were obtained by a silica coating, APS grafting and polymer PBLG covalent modification, with gadolinium phosphate as the centerpiece. Improved dispersion and stability of inorganic micro/nanoparticles in the scaffold while maintaining better MRI tracer properties promoted osteogenic differentiation. The main reasons for analyzing the mechanism of the effect of various modifiers on the osteogenic/chondrogenic induction activity of MC3T3-E1 cells are due to the following reasons.

The first is the effect of particle composition and concentration. At suitable concentrations, it has been demonstrated that gadolinium phosphate promotes OCN expression and promotes new bone regeneration via the Smad/Runx2 pathway [18,19,53]. SiO_2_ promotes collagen expression [49,50,52], APS has better biocompatibility [41] and PBLG promotes osteogenic mineralization [30,33,44]. Different components also cause different ions/groups to be released, further affecting cellular pathways and functions.

Secondly, the different surface groups, charges and pH values of the materials are the ‘first impression’ of the nanoparticles (NPs) to the cell or biomolecule. When solid NPs are introduced into the liquid phase, a different range of repulsive or attractive forces develop between the NPs and other molecules in the environment, and due to the small size and concentrated surface charge, the NPs become highly reactive, facilitating interactions between the NPs and the molecules, which in a physiological environment forms a protein corona that further regulates osteogenic niche capacity by triggering or moderating inflammation [10].

Finally, the introduction of the surface topography of smart materials is also the first clue to cell-aware scaffolds, which control the early biological responses of cells, including adhesion, spreading and migration, and then alter their phenotype, thus controlling the bone regeneration process [54]. The morphology/size/pore structure of nanomaterials such as nanogrooves, nanofibers, nanodots, nanotubes and complex shaped patterns which are nanostructured surfaces have a significant impact on the induction of osteogenic differentiation.

Above all, it is hypothesized that particle composition/concentration, surface groups/charge/pH may have been the main influence on osteogenic differentiation in this research work. The main influences on different research systems are different and therefore the same theory cannot be used to explain different work.

## 3. Materials and Methods

### 3.1. Materials

TEOS (ethyl orthosilicate), isopropanol, ammonia, ethanol, tetrahydrofuran (THF), dioxane and acetone were purchased from Beijing Chemical Works. 3-Aminopropyltriethoxysilane (APS) was purchased from Tokyo Chemical Industry (Tokyo, Japan). g-Benzyl-L-glutamic acid was purchased from Sinopharm Chemical Reagent Co., Technology Co., Ltd. (Shanghai, China). All solvents and chemicals were analytically pure and used directly. Deionized water was used for all aqueous solutions.

### 3.2. Preparation of GdPO_4_·H_2_O@SiO_2_

GdPO_4_·H_2_O nanobunches were prepared according to the previous literature [18]. Then, GdPO_4_·H_2_O was added to a mixed solution (isopropanol:water = 10:1). The mixture was placed in an ultrasonic water bath at 25 °C for 10 min, and 3.5 mL of ammonia was added and stirred for 15 min. TEOS was added dropwise to the above solution in the ratio m_GdPO4·H2O_:V_TEOS_ (mg/mL) = 1:8, 4:5, 1:1, 5:4, 5:3 and 5:2. After stirring for 12 h, the mixture was separated by centrifugation, washed 4 times with deionized water and then dried in an oven at 60 °C for 12 h to obtain SiO_2_-coated GdPO_4_·H_2_O (GdPO_4_·H_2_O@SiO_2_).

### 3.3. Synthesis of GdPO_4_·H_2_O@SiO_2_–APS and BLG-NCA

The APS-modified intermediate (GdPO_4_·H_2_O@SiO_2_–APS) was synthesized based on a published method [30]. 3-Aminopropyltriethoxysilane (APS) was dissolved in 100 mL of ethanol–water mixture (ethanol:water = 9:1) and stirred for 30 min. GdPO_4_·H_2_O@SiO_2_ was then added to the solution in the ratio of GdPO_4_·H_2_O@SiO_2_:APS = 1:0.22 via the sonicating process for 20 min. The mixed solution was adjusted to pH = 10, stirred at room temperature for 6 h, centrifuged three times with absolute ethanol, and dried at room temperature. The APS-modified GdPO_4_·H_2_O@SiO_2_ (GdPO_4_·H_2_O@SiO_2_–APS) was collected after curing for 2 h in a drying oven at 130 °C to strengthen the polysiloxane network structure.

BLG-NCA (L-glutamic acid γ-benzyl ester-N-carbonyl-lactam anhydride), the NCA monomer, was synthesized by the reaction of benzyl glutamic acid with triphosgene in anhydrous tetrahydrofuran (THF) at 50–60 °C.

### 3.4. Synthesis of PBLG-g-GdPO_4_·H_2_O

GdPO_4_·H_2_O@SiO_2_–APS and NCA monomer (GdPO_4_·H_2_O@SiO_2_–APS:NCA = 1:0.125, 1:0.25, 1:0.5 and 1:1) were put into a flame-dried 100 mL round-bottom flask, sealed and then injected with 60 mL of dried dioxane. PBLG-g-GdPO_4_·H_2_O was synthesized after stirring at room temperature for 32 h, centrifugation with dioxane and acetone 5 and 2 times, respectively, and drying at room temperature.

### 3.5. Preparation of Nanocomposites for MR Imaging and Cell Experiments

GdPO_4_·H_2_O, GdPO_4_·H_2_O@SiO_2_, GdPO_4_·H_2_O@SiO_2_–APS and PBLG-g-GdPO_4_·H_2_O were added to N-methylpyrrolidone (NMP) or chloroform solutions with a Gd concentration of 0.4% (*w*/*w*, Gd/PLGA) for MR imaging and cellular experiments, respectively. PLGA was dissolved in the above solutions, dispersed by ultrasound for 10 min and then stirred overnight. For MR imaging, a homogeneous mixture with NMP as the solvent was transferred to a 2 mL syringe and placed in a cryogenic refrigerator for at least 4 h. After replacing the solvent with deionized water to cure for 3 days, the composites were dried at room temperature. In cell experiments, a homogeneous mixture with chloroform as the solvent was dropped onto silica slides and uniformly coated. The composite films of GdPO_4_·H_2_O/PLGA, GdPO_4_·H_2_O@SiO_2_/PLGA, GdPO_4_·H_2_O@SiO_2_–APS/PLGA, and PBLG-g-GdPO_4_·H_2_O/PLGA were dried under vacuum for 48 h. All composites were disinfected by UV light for 2 h prior to cell culture. PLGA was used as a control group.

### 3.6. Characterization of Physical and Chemical Properties

The crystal phase was analyzed by powder X-ray diffraction (XRD; Bruker Co., Bremen, Germany) on a D8 Advance diffractometer using Cu Kα radiation (λ = 0.154 Å). The morphology, structure and size of the samples were determined by field-emission scanning electron microscopy (FESEM; Philips XL30 ESEM FEG, Japan) and transmission electron microscopy (TEM; FEI Tecnai G2 S-Twin, München, Germany). The elemental compositions were analyzed by energy-dispersive X-ray energy spectrometry (EDX; Philips, XL-30 W/TMP, Konan, Japan). Fourier transform infrared spectrometry (FT-IR, Bio-Rad Win-IR Spectrometer, Watford, UK) was recorded in the range of 400–4000 cm^−1^ using the attenuated total reflection (ATR) mode and the KBr slice method. Atom force microscopy (AFM) images were acquired by Bruker’s Dimension Icon and Multimode-V AFM. The amounts of GdPO_4_·H_2_O, GdPO_4_·H_2_O@SiO_2_, GdPO_4_·H_2_O@SiO_2_–APS, and PBLG-g-GdPO_4_·H_2_O were determined by thermogravimetric analysis (TGA, TA Instruments TGA500, New Castle, DE, USA) in air at a heating rate of 10 °C/min from 25 °C to 800 °C.

### 3.7. Magnetic Resonance Imaging In Vitro

MR imaging of GdPO_4_·H_2_O/PLGA, GdPO_4_·H_2_O@SiO_2_/PLGA, GdPO_4_·H_2_O@SiO_2_–APS/PLGA, and PBLG-g-GdPO_4_·H_2_O/PLGA nanocomposites was performed using a 1.5-T scanner (Siemens Magnetom Avanto, Erlangen, Germany). The in vitro imaging parameters were as follows: repeat time TR = 530 ms, echo time TE = 12 ms, field of view = 330 mm × 330 mm, and layer thickness = 2.0 mm.

### 3.8. In Vitro Cell Culture

MC3T3-E1 cells were purchased from the Shanghai Institute of Cell Biology, Chinese Academy of Sciences (CAC) and cultured in regular culture medium containing Dulbecco’s modified Eagle’s medium (DMEM) and 10% v/v fetal bovine serum (FBS) from Gibco (New York, NY, USA) at 37 °C in a humidified atmosphere of 5% CO_2_. The culture medium was refreshed every three days.

Cytotoxicity and proliferation were assessed by the CCK-8 assay. Cytotoxicity was evaluated by using the extracts from the samples. Briefly, GdPO_4_·H_2_O/PLGA, GdPO_4_·H_2_O@SiO_2_/PLGA, GdPO_4_·H_2_O@SiO_2_–APS/PLGA and PBLG-g-GdPO_4_·H_2_O/PLGA films (6 cm^2^/mL) containing 0.4% Gd (*w*/*w*, Gd/PLGA) were immersed in the culture medium for 24 h. MC3T3-E1 cells were cultured at 1 × 10^4^ cells/well for 24 h in 96-well plates. The original medium was then replaced with a fresh 100 μL gradient dilution of the extract, and the cells were incubated under the same conditions for 22 h. The medium was replaced with fresh medium again, 10 μL of CCK-8 was added and incubated for 2 h and the absorbance was measured at 450 nm on a multifunctional microplate scanner (Tecan Infinite M200, Tecan, Männedorf, Switzerland).

For cell adhesion/proliferation experiments, coverslips containing composites were washed three times with PBS and placed in 24-well plates with 4 × 10^4^ MC3T3-E1 cells per well, and cell adhesion was assayed at 6, 12 and 24 h. A total of 1.5 × 10^4^ cells per well were grown for cell proliferation and tested at 1, 3, 7 and 14 days. Cell adhesion and cell morphology were observed by fluorescent isothiocyanate (FITC, Sigma, MO, USA) staining under an inverted microscope (TE2000U, Nikon, Tokyo, Japan). Each subsequent group was prepared in triplicate.

The expression of osteogenic/chondrogenic-related genes was quantitatively assessed by real-time PCR. MC3T3-E1 cells were cultured in 24-well plates containing composite material at 6 × 10^4^ cells per well for 7 and 14 days. The exact procedure can be found in the relevant reference [30].

Calcium-rich deposits of MC3T3-E1 cells were assessed using alizarin red staining in 6-well plates cultured at 3 × 10^4^ cells per well for 3 weeks, following a previously published protocol.

### 3.9. Data Analysis

Quantitative data are expressed as the mean ± standard error. Significance was accepted at * *p* < 0.05.

## 4. Conclusions

In summary, GdPO_4_·H_2_O@SiO_2_, GdPO_4_·H_2_O@SiO_2_–APS and PBLG-g-GdPO_4_·H_2_O were successfully designed and synthesized by controlling the ratio of reactants when _GdPO4·H2O_:V_TEOS_ (mg/mL) = 1:1, GdPO_4_·H_2_O@SiO_2_:APS = 1:0.22 and GdPO_4_·H_2_O@SiO_2_–APS:NCA = 4:1, leading to a grafting ratio of 5.93% for PBLG-g-GdPO_4_·H_2_O, which showed good stability and dispersion in chloroform and the degradable polymer PLGA. After functionalization of the surface modification, only the T_1_-weighted MRI signal of GdPO_4_·H_2_O@SiO_2_ was stronger, and the MRI imaging signals of GdPO_4_·H_2_O@SiO_2_, GdPO_4_·H_2_O@SiO_2_–APS and PBLG-g-GdPO_4_·H_2_O were sequentially weakened. All modification intermediates and the modification product had good biocompatibility, such as an elevated ALP activity to some extent and promoted the expression of COL I and COL II, especially GdPO_4_·H_2_O@SiO_2_ which had significant up-regulation ability on the gene expression of biomacromolecules COL I and COL II, with expression levels at least 4.8 and 3.1 times higher than those of the PLGA group, respectively. The design and synthesis of intelligent and biofunctionalized gadolinium phosphate nanobunches through surface modification will make an outstanding contribution in the field of biomedical materials and bone tissue engineering.

## Data Availability

Not applicable.

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
