# Peer review of "Surface Biofunctionalization of Gadolinium Phosphate Nanobunches for Boosting Osteogenesis/Chondrogenesis Differentiation"

_ijms, 2023, doi:10.3390/ijms24032032_

Round 1

Reviewer 1 Report

Dear Authors,

It is a great pleasure to be recognized as a reviewer of Your manuscript. It deals with a very important and current research topic.

While reading the content of your publication, I had some doubts that I would like to clarify.

The aim of the study is not clearly emphasized in the manuscript.

The character of research seems to be a high advanced material investigation. There is no molecular mechanisms and aspect described in it to present the rationale to put it in the scope of IJMS.

The Results and Discussion chapter is really confusing for me. It is consisted of some material and methods information and have hardly no real discussion with the others researchers as three first subchapters have just 3 references (the third one has none) and are not real discussion but reporting authors own results and opinion.

Even the last subchapter presents references only in big blocks to support results not to discuss. This part of the manuscript refers to the topic of the Journal and should be more emphasized and structured.

This chapter should be very structurally rebuilt.

The Conclusion chapter seems to be a summary of the manuscript, mainly summary of the result chapter. It should be rebuilt and consist of 2-3 sentences that are real conclusions avoiding extensive summary form.

What is the main conclusion for researchers of molecular phenomena from your research?

Best regards

Reviewer 2 Report

The article entitled Surface Biofunctionalization of Gadolinium Phosphate Nanobunches for Boosting Osteogenesis/Chondrogenesis Differentiationis a document of interesting , attractive, and novel subject matter.

However, it needs some major changes before being accepted. Make the following corrections:

1.      'Title' seems very unique as per current research trends. While, 'Abstract' should more focus on main research outcomes and novelty should mention, which is missing. Add 1 or 2 lines as per novelty of work for indicating impact the Surface Biofunctionalization of Gadolinium Phosphate Nanobunches for biomedical applications in 'Abstract' section.

2.      'Introduction' section not enough discussed with biomedical applications of Surface Biofunctionalization of Gadolinium Phosphate Nanobunches  should follow the cited links given:

- https://doi.org/10.1002/btm2.10262

-https://doi.org/10.1016/j.colsurfb.2022.112771

- https://doi.org/10.1021/acsnano.9b04436

- https://doi.org/10.3390/ceramics4040039

- https://doi.org/10.1080/00914037.2020.1848828

3.      In the conclusion, provide a brief explanation about the future perspective of the developed Surface Biofunctionalization of Gadolinium Phosphate Nanobunches and how it can be modified to exhibit better performance.

4.      How likely is the new agent approved by FDA?

5.      Please input magnification for SEM figures (Fig.1 and 2) to more clarify.

  1. Please do more discussion on X-ray diffraction (XRD) of samples..
  2. Please do more discussion regarding  results on Zeta potential in Table 1 for different samples.

As it stands, concern author should give another chance to revise their article and should highlight in the revised manuscript text, so far recommended that the article is mandatory for Major revision'.

Round 2

Reviewer 1 Report

Dear Authors,

Thank you for Your manuscript improvements. 

I still have some doubts about the aim of the paper as it is not highlighted in the body of the manuscript and the conclusions still look like an abstract rather than 1, 2 or maximum 3 short points easy for the Reader to find.

Best regards

Reviewer 2 Report

Authors addressed all comments carefully and completely.
